# Association of Differentially Altered Liver Fibrosis with Deposition of TGFBi in Stabilin-Deficient Mice

**DOI:** 10.3390/ijms241310969

**Published:** 2023-06-30

**Authors:** Jessica Krzistetzko, Cyrill Géraud, Christof Dormann, Anna Riedel, Thomas Leibing

**Affiliations:** 1Department of Dermatology, Venereology, and Allergology, University Medical Center and Medical Faculty Mannheim, Heidelberg University, 68135 Mannheim, Germany; jessica.krzistetzko@medma.uni-heidelberg.de (J.K.);; 2Section of Clinical and Molecular Dermatology, University Medical Center and Medical Faculty Mannheim, Heidelberg University, 68135 Mannheim, Germany; 3European Center for Angioscience (ECAS), Medical Faculty Mannheim, Heidelberg University, 68135 Mannheim, Germany

**Keywords:** liver fibrosis, scavenger receptors, TGFBi

## Abstract

Liver sinusoidal endothelial cells (LSECs) control clearance of Transforming growth factor, beta-induced, 68kDa (TGFBi) and Periostin (POSTN) through scavenger receptors Stabilin-1 (Stab1) and Stabilin-2 (Stab2). Stabilin inhibition can ameliorate atherosclerosis in mouse models, while Stabilin-double-knockout leads to glomerulofibrosis. Fibrotic organ damage may pose a limiting factor in future anti-Stabilin therapies. While Stab1-deficient (*Stab1−/−*) mice were shown to exhibit higher liver fibrosis levels upon challenges, fibrosis susceptibility has not been studied in Stab2-deficient (*Stab2−/−*) mice. Wildtype (WT), *Stab1−/−* and *Stab2−/−* mice were fed experimental diets, and local ligand abundance, hepatic fibrosis, and ligand plasma levels were measured. Hepatic fibrosis was increased in both *Stab1−/−* and *Stab2−/−* at baseline. A pro-fibrotic short Methionine-Choline-deficient (MCD) diet induced slightly increased liver fibrosis in *Stab1−/−* and *Stab2−/−* mice. A Choline-deficient L-amino acid-defined (CDAA) diet induced liver fibrosis of similar distribution and extent in all genotypes (WT, *Stab1−/−* and *Stab2−/−*). A hepatic abundance of Stabilin ligand TGFBi correlated very highly with liver fibrosis levels. In contrast, plasma levels of TGFBi were increased only in *Stab2−/−* mice after the CDAA diet but not the MCD diet, indicating the differential effects of these diets. Here we show that a single Stabilin deficiency of either Stab1 or Stab2 induces mildly increased collagen depositions under homeostatic conditions. Upon experimental dietary challenge, the local abundance of Stabilin ligand TGFBi was differentially altered in Stabilin-deficient mice, indicating differentially affected LSEC scavenger functions. Since anti-Stabilin-directed therapies are in clinical evaluation for the treatment of diseases, these findings bear relevance to treatment with novel anti-Stabilin agents.

## 1. Introduction

Organ-specific vasculature is marked by the heterogeneity of all vascular cell types, with endothelial cells as one of the most characterized populations. LSECs, the microvascular endothelial cells in the liver lobule, are discontinuous endothelial cells lacking a basement membrane [1]. Furthermore, LSECs are professional scavenger endothelial cells since they are able to clear many types of circulating macromolecules in plasma from circulation [2]. Among many receptors responsible for their professional scavenging function, they highly express the scavenger receptor class H group of endocytosis receptors, which consists of Stabilin-1 (Stab1) and Stabilin-2 (Stab2) [3].

Stab1 was first identified in discontinuous sinusoidal endothelial cells and alternatively activated macrophages as “MS-1 high-molecular weight protein” [4,5], while Stab2 was initially named “endocytic hyaluronan (HA) receptor” due to its ability to internalize hyaluronic acid [6]. Later, it was determined that MS-1 and the endocytic HA receptor were homologous to a high degree and belonged to the same protein family [7]. The extracellular domains of Stab1 and Stab2 consist of seven fasciclin, several epidermal growth factor domains, and one HA-binding X-link domain, which is non-functional in Stab1 [7]. Fasciclin domains play a central role in Stabilin-mediated scavenging. Direct Stabilin ligands TGFBi and POSTN, together with Stab1 and Stab2, represent the only four proteins containing fasciclin domains in mammals. Stab1 and Stab2, likely due to fasciclin domain binding, directly bind TGFBi and POSTN; thereby, Stabilin deficiency leads to elevated TGFBi and POSTN plasma levels [8].

Both Stabilin ligands TGFBi and POSTN are part of the extracellular matrix in normal liver tissue [9] and have previously been described as markers in various pathological processes in the liver, including hepatocellular carcinoma. TGFBi is upregulated in human plasma in non-alcoholic fatty liver disease but not in a high-fat diet-induced hepatopathy mouse model [10]. Mice lacking TGFBi were resistant to high-fat diet-induced steatosis in another study [11]. Furthermore, TGFBi expression in the liver and plasma was found to be tightly connected to hepatic collagen levels and turnover rates [12]. TGFBi expression is higher in hepatocellular carcinoma, among many other tumor entities [13].

POSTN was implicated in the development of liver fibrosis [14], whereas POSTN-deficient mice were protected from chronic liver damage in chronic CCl_4_ treatment [15] as well as four weeks of MCD diet [16]. POSTN plasma levels are a potential biomarker of NAFLD in human patients [17]. In HCC, POSTN-deficient mice showed reduced diethylnitrosamine-induced liver cancer [18], while POSTN expression in hepatocellular carcinoma was associated with a poor prognosis in human patients [19,20].

Recently, we have shown that LSECs regulate the molecular composition of the circulating blood and thereby control the homeostasis and functions of distant organs through Stabilin receptors. We generated Stab1-deficient (*Stab1−/−*) and Stab2-deficient (*Stab2−/−*), as well as Stab1/2-double-deficient (Stab-DKO) mice [21]. *Stab1−/−* and *Stab2−/−* mice showed a normal lifespan, while Stab-DKO mice presented with severe glomerular fibrosis, albuminuria, mild perisinusoidal liver fibrosis, and a reduced life expectancy. Kidneys of Stab-DKO mice transplanted into WT mice showed an ameliorated progression of glomerular fibrosis, indicating that glomerulofibrotic nephropathy was probably caused by circulating blood factors in Stab-DKO mice [21]. Age-dependent depositions of direct Stabilin ligands TGFBi and POSTN in kidney and liver tissue of Stab-DKO mice may be detrimental to kidney function; although POSTN knockout from Stab-DKO mice did not show a completely rescued kidney function or lifespan, a slightly improved glomerular phenotype was observed [22]. Recently, we were able to show that genetic single Stabilin deficiency and anti-Stabilin antibody therapy in different atherosclerosis-prone mouse models ameliorate atherosclerosis, likely through altered immune cell activation mediated by a plasma proteome switch including TGFBi and POSTN [8].

Liver cirrhosis is a late stage of different chronic liver pathologies, ultimately leading to hepatocellular carcinoma. Defenestration and capillarization of LSECs represent an important step in the progression of liver fibrosis and, ultimately, cirrhosis [23]. Stab-DKO leads to non-compensable transcriptomic alterations of LSEC-specific functional genes that potentiate when both Stabilins are missing [24]. Interestingly, Stab1 deficiency aggravates liver fibrosis upon chronic liver injury using CCl_4_ and six weeks of MCD diet as models [25]. The involvement of both Stab1 and Stab2 in the development and progression of non-alcoholic steatohepatitis (NASH) models has not been comprehensively studied. The aim of this study was to characterize the effect of single Stabilin deficiency on different preclinical liver fibrosis models and how these models affect ligand clearance and deposition.

## 2. Results

### 2.1. Short MCD Diet Leads to Slight Hepatic Steatosis and Mild Fibrosis in All Genotypes

H&E stainings and Oil Red O stainings showed an increase in lipid droplets presenting as microvesicular steatosis in mouse liver sections of all genotypes after the MCD diet. Apart from that, no overt morphological alterations were observed (Figure 1A–C). Mice showed marked weight loss after one week of the MCD diet. After two weeks, a weight loss of 20% was observed (Appendix A). The liver weight showed a very strong decrease in all genotypes, as expected (Appendix A). Sirius Red staining of liver tissue (Figure 1D) demonstrated that *Stab1−/−* and *Stab2−/−* deficient mice fed a chow diet showed increased collagen fiber deposition compared to chow-fed WT mice. After MCD treatment, all genotypes showed an increased signal in liver collagen stainings with the highest levels in *Stab1−/−*, followed by *Stab2−/−* and the lowest signal in WT mice (Figure 1E), the same order as in chow-fed mice. This phenomenon was confirmed by a collagen assay performed on liver tissue (Figure 1F). Mouse plasma analysis revealed that MCD-treated mice showed a decrease in glucose, triglycerides, and cholesterol levels (Appendix A). Plasma Cholinesterase and enzyme levels of ALT, AST, and GLDH were significantly increased in all genotypes after MCD treatment (Appendix A).

### 2.2. CDAA-Induced Fibrosis Is Similar in Stabilin-Deficient Mice Compared to Wildtype Mice

H&E and Oil Red O stainings of liver sections from CDAA-treated mice showed strong macrovesicular steatosis, which did not differ between genotypes (Figure 2A–C). Sirius Red staining of liver tissue demonstrated higher collagen abundance in chow-fed *Stab1−/−* and *Stab2−/−* mice compared to Wildtype. After one week of the CDAA diet, mice showed a slight weight loss in comparison to chow-fed mice, but after two weeks of treatment, the weight stabilized and developed normally (Appendix A). After ten weeks of the CDAA diet, livers showed changes in color and size (Appendix A), as expected. Liver and spleen weight showed a very strong increase (Appendix A) compared to livers harvested from mice on a chow diet. Interestingly, after 10 weeks of CDAA treatment, all genotypes showed hepatic fibrosis of similar pattern and extent (Figure 2D,E), which was again confirmed with a collagen assay performed on liver tissue (Figure 2F).

### 2.3. No Stellate Cell Activation upon Experimental Diets

Immunofluorescent (IF) stainings of stellate cell markers α-SMA and Desmin and in situ hybridization of platelet-derived growth factor receptor beta (PDGFRB) did not show changes in MCD fed mice regardless of genotype compared to chow-fed controls (Appendix A–F). Immunohistochemical stainings of α-SMA and Desmin in livers from CDAA-treated mice revealed no gross differences (Appendix A). As α-SMA staining was only found in perivascular smooth muscle cells/pericytes, there was no indication of increased activation of stellate cells in all models.

### 2.4. A Sinusoidal LSEC-Phenotype Was Mostly Preserved upon MCD and CDAA Diet

Capillarization of LSEC is commonly observed in liver disease and is well known to be accompanied by the downregulation of LSEC-characteristic transmembrane proteins such as CD32b and Lyve-1 and the upregulation of proteins characteristic of continuous EC such as Endomucin (Emcn). Immunofluorescent stainings of CD32b, Lyve-1, and Emcn did not show differences in distribution or intensity between control mice and MCD-treated mice (Figure 3C–H). Immunofluorescent staining of Intercellular Adhesion Molecule 1 (ICAM-1) showed a very strong and significant decrease in ICAM-1 signal after two weeks of MCD treatment in all genotypes (Figure 3A,B). In CDAA-treated mice, all stainings were performed using immunohistochemical stainings because of the strong background signal in immunofluorescent stainings due to marked steatosis. No overt differences between the genotypes were observed (Figure 4A–C). CD32b IHC staining was slightly more abundant in *Stab2−/−* mice compared to *Stab1−/−* mice, which reached significance after CDAA treatment (Figure 4D,E). 

### 2.5. Experimental Diets Do Not Induce Robust Changes in Liver POSTN Abundance and POSTN Plasma Levels

Stainings for POSTN revealed no POSTN-positive structures in all liver sections, with glomeruli from Stab-DKO mice serving as a positive control (Appendix A). Simple Western revealed that chow-fed, *Stab2−/−* mice presented a higher liver POSTN level than WT mice on a chow diet, and after MCD treatment, *Stab1−/−* showed a significantly higher POSTN level than WT and *Stab2−/−* mice (Appendix A). POSTN ELISA of plasma revealed higher POSTN levels in chow-fed *Stab1−/−* mice in comparison to WT mice. After MCD treatment, no significant changes or differences between the genotypes were observed (Appendix A). In CDAA-treated mice, Simple Western revealed a trend toward an increase in POSTN level in the liver after the CDAA diet; only WT mice reached significance (Appendix A). POSTN ELISA of plasma showed a significantly higher level of POSTN in *Stab1−/−* chow-fed mice in comparison to *Stab2−/−* chow-fed mice. After CDAA treatment, WT and *Stab1−/−* mice showed a significant decrease in POSTN level in comparison to chow-fed mice of these genotypes. *Stab2−/−* mice did not show any changes (Appendix A).

### 2.6. Hepatic and Circulating Abundance of TGFBi Are Differentially Altered in Stab-Deficient Mice by MCD and CDAA Diet

MCD-fed mice generally showed higher liver TGFBi levels, which were confirmed using immunofluorescent stainings (Figure 5A,B) and Simple Western (Figure 5C,D). After treatment, *Stab1−/−* mice showed the strongest increase in TGFBi immunofluorescent signal and also the highest level in comparison to WT and *Stab2−/−* mice (Figure 5A,B). Simple Western analyses of TGFBi from liver protein confirmed MCD-fed *Stab1−/−* mice showed the highest TGFBi level (Figure 5C,D). Furthermore, we performed TGFBi ELISA from plasma in our models. Chow-fed MCD mice showed the highest TGFBi plasma levels, and the MCD diet did not significantly increase TGFBi plasma levels in any model. In CDAA-treated mice, immunohistochemical staining of TGFBi in the liver showed a significant increase in MCD-fed WT mice compared to chow and a trend toward increased TGFBi abundance in Stabilin-deficient mice (Figure 6A,B). Simple Western of TGFBi from liver protein (Figure 6C,D) showed that all genotypes had similar levels of TGFBi after treatment with CDAA. TGFBi ELISA from plasma samples revealed a small increase in the CDAA diet, which was only significant in *Stab2−/−* mice. In chow-fed mice, *Stab1−/−* mice showed the highest levels, which is in line with the results from the MCD control group (Figure 5E and Figure 6E).

### 2.7. Hepatic TGFBi and POSTN Correlate with Liver Fibrosis Level

Since TGFBi abundance in the liver showed a similar distribution to fibrosis levels, we correlated hepatic TGFBi levels with fibrosis levels assessed by Sirius Red stainings. Here, we found a very high correlation of TGFBi with liver fibrosis in all pooled experiments with pooled genotypes and treatments (R^2^ = 0.563, r = 0.75, *p* < 0.0001) (Figure 7A). Furthermore, we correlated POSTN liver abundance with fibrosis and found a moderate to high correlation (R^2^ = 0.244, r = 0.49, *p* < 0.0001) (Figure 7B). To further investigate the differential effects of feeding and genotype on ligand levels in the liver, we also separated genotypes and re-analyzed the correlation between fibrosis and ligand abundance. Here, we were able to show that over all experiments, the correlation coefficient for TGFBi and fibrosis was very high for *Stab2−/−* mice (R^2^ = 0.82, r = 0.91, *p* < 0.0001) and WT mice (R^2^ = 0.59, r = 0.77, *p* = 0.0009) and high for *Stab1−/−* mice (R^2^ = 0.42, r = 0.65, *p* = 0.0006) (Figure 7C). For POSTN, WT mice showed a very high correlation with liver fibrosis (R^2^ = 0.58, r = 0.76, *p* < 0.0001), while *Stab1−/−* showed a moderate to high correlation (R^2^ = 0.24, r = 0.49, *p* = 0.0162) and in *Stab2−/−*, POSTN levels did not correlate with liver fibrosis levels (*p* = 0.2193) (Figure 7D). Finally, we found a high correlation of TGFBi and POSTN levels in all pooled experiments with pooled genotypes and treatments (R^2^ = 0.266, r = 0.52, *p* < 0.0001) (Figure 7E), which was significant in all genotypes except *Stab2−/−*, which showed a trend toward a moderate correlation (R^2^ = 0.13, r = 0.36, *p* = 0.0734) (Figure 7F).

## 3. Discussion

Here, we characterize phenotypic features and dynamics of Stabilin ligands in two diet-induced murine models of NASH-like fibrotic liver disease, extending previous data on *Stab1−/−* mice [21,25]. Using *Stab1−/−* and *Stab2−/−* mice, we can show here that *Stab2−/−* mice show slightly increased fibrosis on a normal chow diet, which is aggravated by the MCD diet. Surprisingly, the CDAA diet induced similar fibrosis in all genotypes, indicating differentially altered susceptibility to these two models in Stabilin-deficient mice. Markers of LSEC-differentiation were not strongly altered between genotypes upon experimental diets, indicating that single Stabilin-deficiency did not induce overt capillarization in these models [21,24]. Stabilin ligand POSTN only correlated with the degree of liver fibrosis in WT and *Stab1−/−* mice across all experiments, while TGFBi, as a structurally similar ligand of both Stabilins, showed a very strong correlation with the degree of fibrosis in all genotypes in liver tissue.

While it was previously known that capillarization of LSECs precedes liver fibrosis [23], less is known about the pleiotropic effects caused by possibly inefficient scavenging by LSECs on liver fibrosis progression and, ultimately, the development of hepatocellular carcinoma. The MCD diet for 4 weeks had only minor effects on Stab2 in LSECs, but not on Stab1 expression [26], while the CDAA diet for 10 weeks moderately decreased Stab1 and Stab2 expression in LSECs [27]. Hyaluronic acid levels were moderately increased after 10 weeks of CDAA feeding, indicating only a mild effect on scavenging by Stab2 [27] since Stab1 is not involved in hyaluronic acid scavenging.

Since our previous studies have shown that Stab1 and Stab2 can compensate each other to a certain degree, regarding scavenging of ligands like POSTN and TGFBi [8,22], we tested whether a pro-fibrotic stimulus adding on single Stabilin deficiency might trigger pronounced liver fibrosis and ligand deposition comparable to Stabilin-double-deficient mice. Interestingly, only a moderate pro-fibrotic stimulus with a MCD diet for two weeks showed differences between genotypes regarding overall fibrosis, which strongly mirrored baseline changes in fibrotic burden observed in each genotype. A CDAA diet for 10 weeks, which by comparison induced much stronger fibrosis, did not reveal a specific susceptibility to stronger liver fibrosis in single-Stabilin-deficient animals. A very high correlation of TGFBi to overall fibrotic burden was observed in the liver, indicating that loss of Stab1 or Stab2 alone does not seem to impede scavenging of POSTN or TGFBi, even in liver disease models in these mice.

Since TGFBi levels in the liver correlate with fibrosis in all our models and genotypes here, we hypothesize that the TGFBi protein in the liver we observed is not deposited as we previously hypothesized for high TGFBi and POSTN levels in glomeruli and liver tissue from Stab-DKO tissue [22]. Since no immunofluorescent signal was observed for POSTN, even in strongly fibrotic CDAA-fed livers, we hypothesize that POSTN positivity in Stab-DKO livers is indeed due to deposition from the plasma and fibrogenic stimuli alone are not sufficient to induce strong POSTN depositions as seen in Stab-DKO [22]. Correlation for POSTN and fibrosis was overall weaker than for TGFBi and fibrosis and did not reach significance in *Stab2−/−* mice, although TGFBi and POSTN levels correlated well overall, which is in line with previous findings [22]. Since TGFBi and POSTN plasma levels were only moderately increased in Stabilin-deficient mice, we did not observe increased plasma levels upon experimental diets; a likely explanation is a compensation of the other Stabilin upon single Stabilin-deficiency, even upon reduced Stabilin expression levels previously observed in CDAA treatment [27].

Our findings here give further insights into the scavenging of fasciclin domain proteins TGFBi and POSTN, which are heavily implicated in liver fibrosis and hepatocellular carcinoma, through Stab1 and Stab2 and in liver fibrosis models. Stabilin single-deficiency does not increase fibrosis-associated TGFBi levels in the liver upon experimental diets and in homeostasis, while Stabilin double-deficiency leads to age-dependant TGFBi and POSTN depositions in the kidney and liver under homeostatic conditions [22]. Since anti-Stab1 therapies are already in Phase I/II clinical trials [28] and Stab1/Stab2 inhibition has therapeutic promise in atherosclerosis [8], non-homeostatic disease models of genetic Stabilin deficiency might allow predicting side-effects of anti-Stabilin antibodies.

To conclude, we show how liver fibrosis in preclinical models differentially correlates with TGFBi and POSTN levels in response to deficiency for Stab1 or Stab2, bearing consequences for potential anti-Stabilin therapies.

## 4. Materials and Methods

### 4.1. Animal Studies

Animals were held in a 12-h day/night cycle under specific pathogen-free (SPF) conditions and fed ad libitum. To investigate the development of liver damage and inflammation in different metabolic liver fibrosis models, male mice on the C57BL/6J background with different genotypes (WT, *Stab1−/−* and *Stab2−/−*) [21] were fed with the MCD diet (ssniff E15653-94) for 2 weeks or with a CDAA diet (ssniff E15666-94) for 10 weeks to induce liver fibrosis. Control groups were fed with a normal chow diet (ssniff V1534-000). Blood samples were collected at different time points, and organs were harvested at the end of the experiment.

### 4.2. Plasma Analysis

Plasma levels of glucose, triglycerides, cholesterol, cholinesterase, and the liver enzymes ALT, AST, and GLDH were analyzed using a Roche Cobas C311 Chemistry Analyzer.

### 4.3. Histopathological Analysis

For Hematoxylin & Eosin (H&E) and Sirius Red staining, formalin-fixed, paraffin-embedded samples were processed according to standard protocols. Sirius Red signal was quantified using channel separation and thresholding using the software ImageJ [29].

### 4.4. Oil Red O Staining

Freshly cut frozen liver tissue (8 µm) was dried for 30 min and then immersed in ddH_2_0. The tissue was then incubated in 60% Isopropanol followed by staining for 10 min in fresh filtered Oil Red O solution (stock solution: 0.5 g Oil Red O in 99% 2-Propanol, working solution: 6 parts of stock solution with 4 parts of ddH_2_0). After that, tissue was differentiated in 60% Isopropanol and washed in ddH_2_0. The last step was the incubation of the tissue in a hematoxylin solution for nuclear staining.

### 4.5. Immunofluorescence Stainings

Paraffin sections (3–4 µm) were prepared according to standard protocols; antigen retrieval was performed with HIER (heat-induced-epitope-retrieval) buffer (pH 6.0) in a 95 °C water bath for 30 min. The primary antibody (Table 1) was diluted in antibody diluent (Dako) and was incubated overnight at 4 °C; the fluorescent secondary antibody was incubated for 1 h at room temperature in combination with DAPI in antibody dilution (Dako).

### 4.6. IHC Stainings

Sectioned tissue was prepared as described above. After antigen retrieval, the tissue was blocked for 10 min with Peroxidase Blocking Buffer. First, antibodies were incubated overnight at 4 °C. Subsequently, HRP-coupled secondary antibody was incubated for 1 h at room temperature. After a washing step, the sections were incubated with AEC substrate chromogen for 5–30 min at room temperature. After that, nuclear staining with hematoxylin solution was performed.

### 4.7. In Situ Hybridization

For in situ hybridization, an RNAscope 2.5 HD Detection Kit (RED) with standard probes (Table 2) was used (ACDbio).

### 4.8. ELISA Assays

For both ELISA assays (TGFBi and POSTN), mouse Plasma was used. Both ELISAs were performed according to the manufacturer’s specifications. For TGFBi ELISA, plasma was diluted 1:500, and for POSTN ELISA, plasma was diluted 1:2000. The optical density was measured with a TECAN Microplate-Reader at the wavelength of 450 nm.

### 4.9. Collagen Assay

For measuring the collagen level in liver tissue, the sensitive Tissue Collagen Assay Kit from QuickZyme (QZBtiscol5) was used. Frozen liver tissue, stored at −80 °C, was thawed before starting the assay. The assay was performed to the manufacturer’s specifications. The optical density was measured with a TECAN Microplate-Reader at the wavelength of 570 nm.

### 4.10. Simple Western

For analyzing the concentrations of TGFBi and POSTN in liver tissue lysates, Simple Western (JESS, BioTechne) was used with a protein normalization kit. For titrating the antibodies and the amount of tissue, the separation module kit (12–230 kDa Separation; SM-W001) from BioTechne was used and was performed to the manufacturer’s specifications. After titrating, the protein normalization module kit (Protein Normalization Separation 12–230 kDa; SM-PN01, Bio-Techne) was used and performed to the manufacturer’s specifications.

### 4.11. Statistics

The results of the quantification of the stainings were analyzed with the software Graph Pad Prism, Version 9. Two separated two-way ANOVAs were performed to compare either all genotypes within one treatment or to compare two treatments of one genotype. To show significance, a Šidák-correction was performed. * show significances between treatments and # show significances between genotypes. The symbols show the following significances: *p* ≤ 0.05 = */#; *p* < 0.01 = **/##; *p* < 0.001 = ***/###; *p* < 0.0001 = ****/####; n.s. means not significant. The results of correlation analysis were analyzed with the software Jmp and were valued according to the correlation coefficient (r) of Pearson and the significance (*p*-value). A value of r < 0.01 shows no correlation, 0.1 < r > 0.3 shows a small correlation, 0.3 < r > 0.5 shows a moderate correlation, 0.5 < r > 0.7 shows a strong correlation, and r > 0.7 shows a very strong correlation. The significance of p-values is described above.

## Figures and Tables

**Figure 1 ijms-24-10969-f001:**
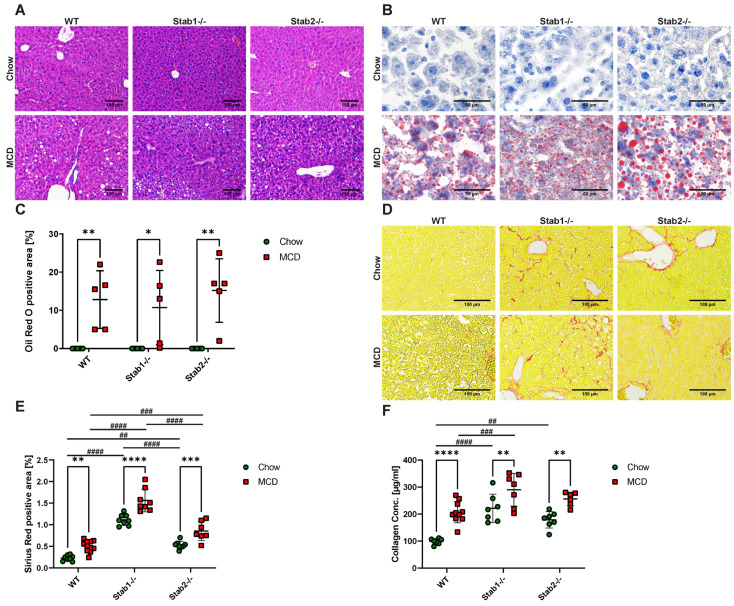
Effects of MCD diet on hepatic steatosis and fibrosis. (**A**) H&E staining of representative liver sections (scale bar = 100 µm). (**B**) Oil Red O staining of representative liver sections (scale bar = 50 µm). (**C**) Quantification of Oil Red O positive staining. (**D**) Sirius Red staining of representative liver sections (scale bar = 100 µm). (**E**) Quantification of Sirius Red positive staining. (**F**) Collagen assay of liver tissue of control mice and MCD mice. * show significances between treatments and # show significances between genotypes. The symbols show the following significances: *p* ≤ 0.05 = *; *p* < 0.01 = **/##; *p* < 0.001 = ***/###; *p* < 0.0001 = ****/####; n.s. means not significant.

**Figure 2 ijms-24-10969-f002:**
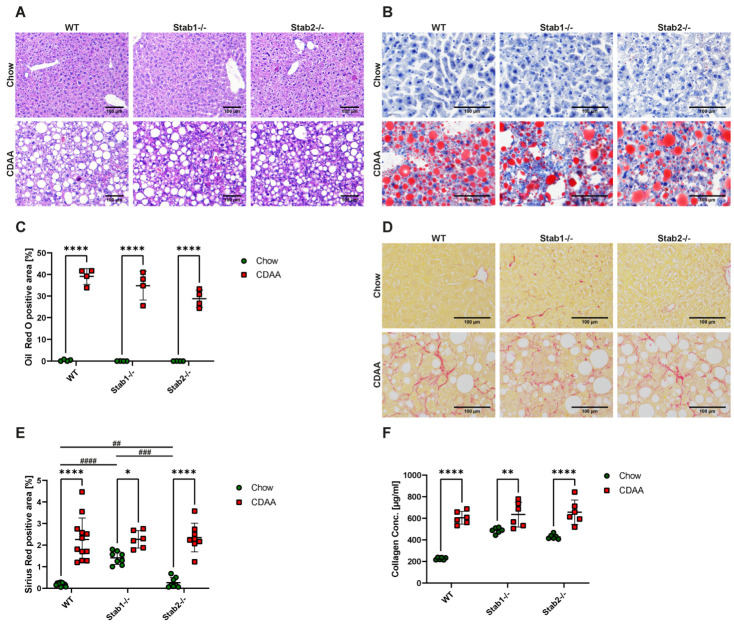
Effects of CDAA diet on hepatic steatosis and fibrosis. (**A**) H&E staining of representative liver sections (scale bar = 100 µm). (**B**) Oil Red O staining of representative liver sections (scale bar = 100 µm). (**C**) Quantification of Oil Red O positive staining. (**D**) Sirius Red staining of representative liver sections (scale bar = 100 µm). (**E**) Quantification of Sirius Red positive staining. (**F**) Collagen assay of liver tissue of control mice and CDAA mice. * show significances between treatments and # show significances between genotypes. The symbols show the following significances: *p* ≤ 0.05 = *; *p* < 0.01 = **/##; *p* < 0.001 = ###; *p* < 0.0001 = ****/####; n.s. means not significant.

**Figure 3 ijms-24-10969-f003:**
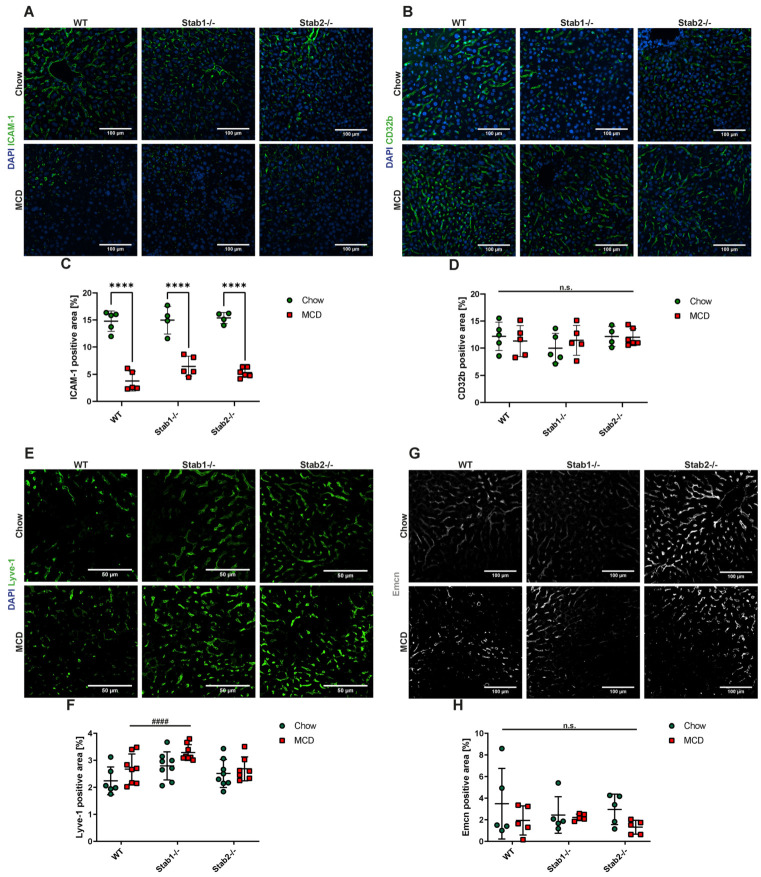
Effects of MCD diet on liver endothelial cells. (**A**,**B**) IF staining and quantification of ICAM-1 positive staining (scale bar = 100 µm). (**C**,**D**) IF staining and quantification of CD32b positive staining (scale bar = 100 µm). (**E**,**F**) IF staining and quantification of Lyve-1 positive staining (scale bar = 50 µm). (**G**,**H**) IF staining and quantification of Emcn positive staining (scale bar = 100 µm). * show significances between treatments and # show significances between genotypes. The symbols show the following significances: *p* < 0.0001 = ****/####; n.s. means not significant.

**Figure 4 ijms-24-10969-f004:**
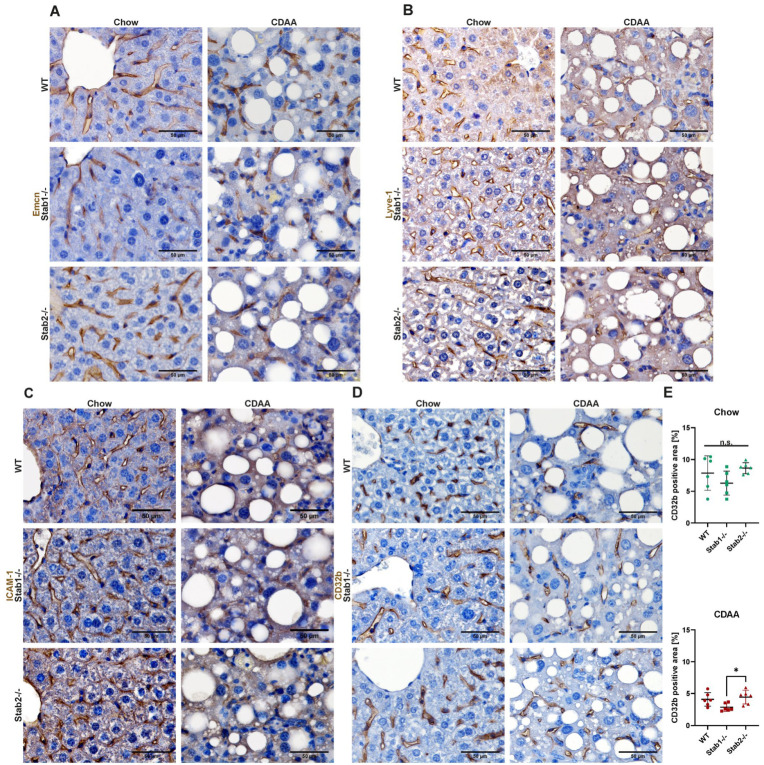
Effects of CDAA diet on liver endothelial cells. (**A**) IHC staining of Emcn (scale bar = 50 µm). (**B**) IHC staining of Lyve-1 (scale bar = 50 µm). (**C**) IHC staining of ICAM-1 (scale bar = 50 µm). (**D**) IHC staining of CD32b (scale bar = 50 µm). (**E**) Quantification of CD32b positive staining, separated to chow (upper panel) and CDAA treated (lower panel) mice. * show significances between treatments. The symbols show the following significances: *p* ≤ 0.05 = *; n.s. means not significant.

**Figure 5 ijms-24-10969-f005:**
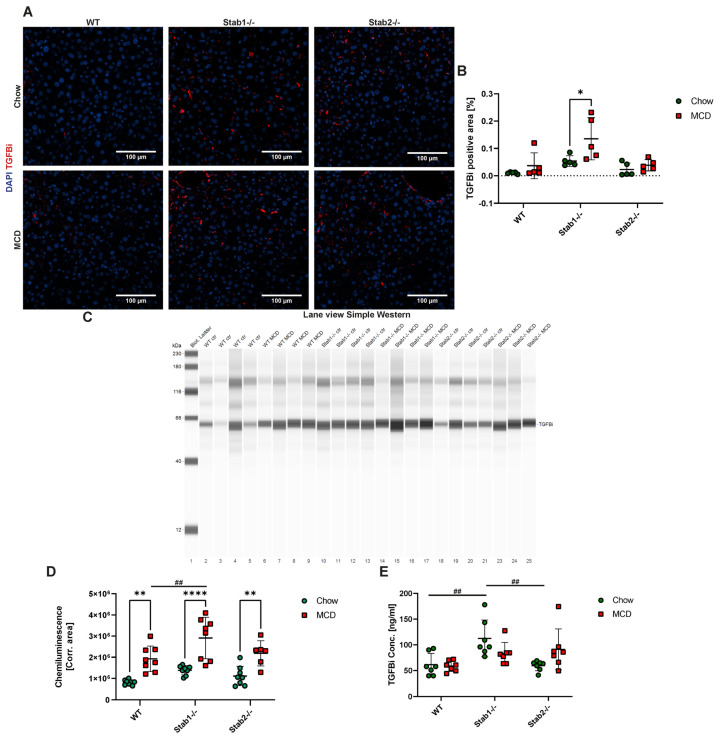
Effects of MCD diet on Stabilin ligand TGFBi. (**A**) TGFBi IF staining of representative liver sections (scale bar = 100 µm). (**B**) Quantification of TGFBi positive staining. (**C**) Lane view of TGFBi Simple Western from liver protein. (**D**) Quantification of TGFBi Simple Western. (**E**) Quantification of TGFBi ELISA from plasma. * show significances between treatments and # show significances between genotypes. The symbols show the following significances: *p* ≤ 0.05 = *; *p* < 0.01 = **/##; *p* < 0.0001 = ****; n.s. means not significant.

**Figure 6 ijms-24-10969-f006:**
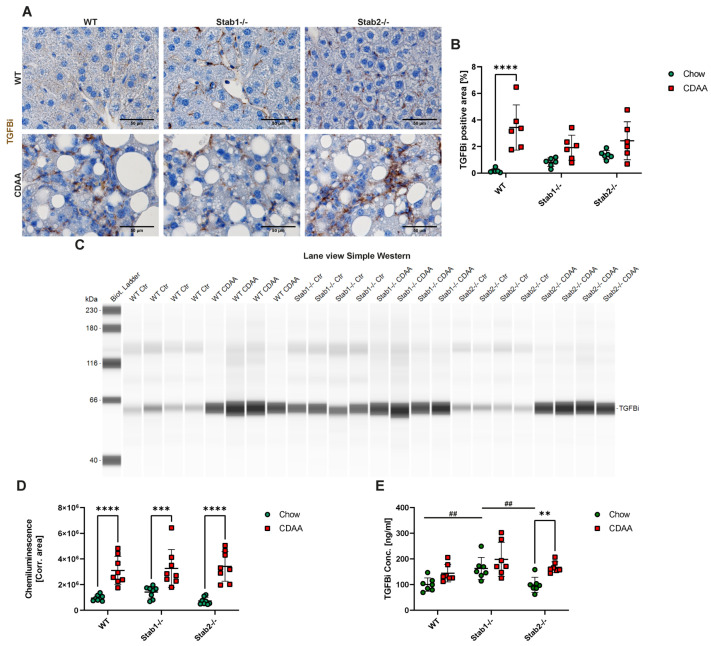
Effects of CDAA diet on Stabilin ligand TGFBi. (**A**) TGFBi IF staining of representative liver sections (scale bar = 100 µm). (**B**) Quantification of TGFBi positive staining. (**C**) Lane view of TGFBi Simple Western from liver protein. (**D**) Quantification of TGFBi Simple Western. (**E**) Quantification of TGFBi ELISA from plasma. * show significances between treatments and # show significances between genotypes. The symbols show the following significances: *p* < 0.01 = **/##; *p* < 0.001 = ***; *p* < 0.0001 = ****; n.s. means not significant.

**Figure 7 ijms-24-10969-f007:**
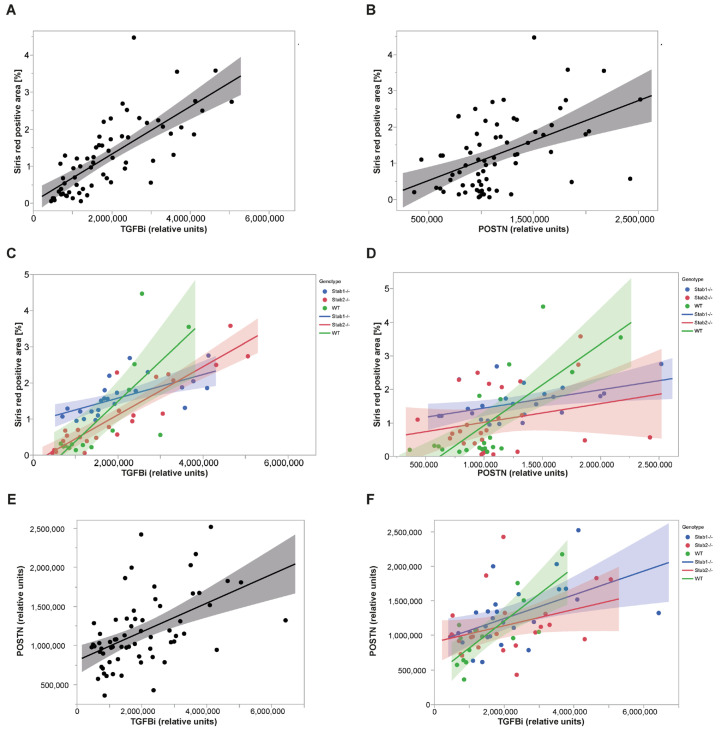
Correlation of Stabilin ligands TGFBi, POSTN, and liver fibrosis. (**A**) Overall correlation of TGFBi levels (Simple Western) and Sirius Red positive area across all genotypes and experiments. (**B**) Overall correlation of POSTN levels (Simple Western) and Sirius Red positive area across all genotypes and experiments. (**C**) Overall correlation of TGFBi levels (Simple Western) and Sirius Red positive area separated by genotypes. (**D**) Overall correlation of POSTN levels (Simple Western) and Sirius Red positive area separated by genotypes. (**E**) Correlation of POSTN levels (Simple Western) and TGFBi levels (Simple Western) across all genotypes and experiments. (**F**) Correlation of POSTN levels (Simple Western) and TGFBi levels (Simple Western) separated by genotypes.

**Table 1 ijms-24-10969-t001:** Antibody list.

Target	Vendor, Order Number	Application
ICAM-1	Proteintech, 10020-1-AP	IHC, IF
CD32b	BioTechne, AF1460	IHC, IF
Lyve-1	ReliaTech, 103-PA50	IHC, IF
Emcn	ThermoFisher, 14-5851-85	IHC, IF
TGFBi	Abcam, ab170874	IHC, IF, Simple Western
POSTN	BioTechne, AF2955	IHC, IF, Simple Western
α-SMA	Abcam, ab5694	IHC, IF
Desmin	Abcam, ab15200	IHC, IF
F4/80	BioLegend, 123102	IHC, IF
CD68	Abcam, ab125212	IHC, IF
CD11b	BioLegend, 101202	IHC, IF

**Table 2 ijms-24-10969-t002:** Probe list used for in situ hybridization.

Target	Vendor	Catalog Number
PPIB	ACDbio	313,911
DapB	ACDbio	310,043
PDGFRB	ACDbio	424,651

## Data Availability

The data that support the findings of this study are available from the corresponding author upon reasonable request.

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
