# Peer review of "Association of Differentially Altered Liver Fibrosis with Deposition of TGFBi in Stabilin-Deficient Mice"

_ijms, 2023, doi:10.3390/ijms241310969_

Round 1

Reviewer 1 Report

Krzistertzko J, et al. reported that experimental liver fibrosis is associated with deposition of TGFBi and is differentially altered in Stabilin-deficient mice during homeostasis and diet-induced mouse models.

1.      In abstract section, in the first, authors should explain about “TGFBi”, “POSTN”, “Stab1” and “Stab2”.

2.      Do not use the abbreviations when they first appear: TGFBi, POSTN, MCD, CDAA, LSEC…

3.      What does it mean “all genotypes”?

4.      In Introduction section, what are MS-1, TGFBi, POSTN, MCD..

5.      In Results section, do not use “H&E”.

6.      Wildtype?

7.      PDGFRB?

8.       ICAM1?

9.       Do not use “Figure 5c”

10.   Do not use Figure 6c.

Authors should ask native English speaker to edit their manuscript.

Author Response

We thank the reviewer for the valuable suggestions. Please find our answers below in response the the italic points.

1. In abstract section, in the first, authors should explain about “TGFBi”, “POSTN”, “Stab1” and “Stab2”.

Since one major concern of this reviewer is the use of abbreviations and language, we thoroughly revised phrasing and abbreviations in the entire manuscript. Since our Abstract is already relatively long and we have to adhere to the journals limtations regarding word count, we would like to refer the reviewer to lines 38-53 for an introduction to Stabilin receptors and to lines 53-71 for an introduction of TGFBi and POSTN.

2. Do not use the abbreviations when they first appear: TGFBi, POSTN, MCD, CDAA, LSEC…

We introduced the abbreviations and included a list of Abbreviations to help readability.

3. What does it mean “all genotypes”?

We included an explanations whenever this phrase is mentioned. Mice genotypes on this paper include Wildtype, Stab1-deficient and Stab2-deficient C57BL/6J mice.

4. In Introduction section, what are MS-1, TGFBi, POSTN, MCD..

As previously mentioned in our answer to point 2., we introduced the abbreviations and included a list of Abbreviations to help readability.

5. In Results section, do not use “H&E”.

We are uncertain about the phrasing of this suggestion. We now refer to "H&E" as H&E staining. We feel that routine histology of experimental models is important and should not be omitted in our manuscript.

6. Wildtype?

Wildtype in our manuscript means standard C57BL/6J mice opposed to Stabilin-deficient mice. We updated lines 311-313 as follows: "To investigate the development of liver damage and inflammation in different metabolic liver fibrosis models, male mice on the C57BL/6J backgorund with different genotypes (WT, Stab1-/- and Stab2-/-)... "

7. PDGFRB?

As previously mentioned in our answer to point 2., we introduced the abbreviations and included a list of Abbreviations to help readability.

8. ICAM1?

As previously mentioned in our answer to point 2., we introduced the abbreviations and included a list of Abbreviations to help readability.

9. Do not use “Figure 5c”

10. Do not use Figure 6c.

We feel that a presentation of interpolated bands, similar to Western blotting, improves confidence of readers in our SimpleWestern results, thereby we would like to keep those figures.

We hope that we addressed all issues accordingly and our manuscript is now acceptable for publication in the International Journal of Molecular Sciences.

Reviewer 2 Report

The mauscript „Experimental liver fibrosis is associated with deposition of 2TGFBi and is differentially altered in Stabilin-deficient mice 3during homeostasis and diet-induced models” is well written and organized. It is suitable for the International Journal of Molecular Sciences. However few issues should be addressed before being accepted for publication.

The title is long and some changes may improve it. For instance, “Experimental liver fibrosis association with TGFBi deposition and is differentially alteration in Stabilin-deficient mice” may be more suitable.

Lines 92-94 “The major aim of this study was to characterize the effect of single Stabilin deficiency on different preclinical liver fibrosis models and how these models affect ligand clearance and deposition.” – I do not see the need to define the aim as major. What is minor than? Just “the aim of this study…” should be enough.  

After line 299 some kind of conclusion is needed. It may be just separate paragraph in which the authors would explain whether they have achieved the aim of the study based on the obtained results without referencing to other papers and solely focusing on this research.

Line 305 “…ground with different genotypes (WT, Stab1-/- and Stab2-/-) (Schledzewski et al., 2011)”- the reference should be given with number as throughout the paper. 

Author Response

We thank the reviewer for the positive evaluation and very valuable suggestions which we adopted in our revised manuscript. Please find our answers below in response the the italic points.

The mauscript „Experimental liver fibrosis is associated with deposition of 2TGFBi and is differentially altered in Stabilin-deficient mice 3during homeostasis and diet-induced models” is well written and organized. It is suitable for the International Journal of Molecular Sciences. However few issues should be addressed before being accepted for publication.

The title is long and some changes may improve it. For instance, “Experimental liver fibrosis association with TGFBi deposition and is differentially alteration in Stabilin-deficient mice” may be more suitable.

The reviewer is correct. We changed the title to "Association of differentially altered liver fibrosis with deposition of TGFBi in Stabilin-deficient mice" and feel it has improved considerably.

Lines 92-94 “The major aim of this study was to characterize the effect of single Stabilin deficiency on different preclinical liver fibrosis models and how these models affect ligand clearance and deposition.” – I do not see the need to define the aim as major. What is minor than? Just “the aim of this study…” should be enough.

We omitted the word "major".

After line 299 some kind of conclusion is needed. It may be just separate paragraph in which the authors would explain whether they have achieved the aim of the study based on the obtained results without referencing to other papers and solely focusing on this research.

We added the following conclusion, lines 308-310 in the revised manuscript: "To conclude, we here show how liver fibrosis in preclinical models differentially correlates with TGFBi and POSTN levels in response to deficiency for Stab1 or Stab2, bearing consequences for potential anti-Stabilin therapies."

Line 305 “…ground with different genotypes (WT, Stab1-/- and Stab2-/-) (Schledzewski et al., 2011)”- the reference should be given with number as throughout the paper.

We apologize, this reference has been updated accordingly.

We hope that we addressed all points accordingly and our manuscript is now acceptable for publication in the International Journal of Molecular Sciences.

Round 2

Reviewer 1 Report

Authors should demonstrate the western blot without cut and paste in Figure 5c and 6c. Authors should show the whole blots.

N/A.

Author Response

We thank the reviewer for reconsidering our manuscript.

As requested, we now show uncropped pictures of the Simple Western lane view in Figures 5c and 6c as well as Fig. S5C and Fig. S6C.

We feel we comprehensively adressed all issues raised by the reviewer.

Round 3

Reviewer 1 Report

Authors did not make revision Figures S5c and S6c.

Unfortunately, this manuscript has a scientific problem.

Authors showed only cropped pictures of the Simple Western lane view in Figures 5c and 6c as well as Fig. S5C and Fig. S6C.

Author Response

We are sorry for the confusion.

We did not perform Western Blotting, but Simple Western, which we clearly stated in the first and subsequent versions of the manuscript:

https://link.springer.com/protocol/10.1007/978-1-4939-2694-7_47

"Lane view" is the name of the rendering of a Western-Blot gel like photomicrograph using proprietary Simple Western technology.

We attached all experiments as files for "Compass" a proprietary but free to use software (https://www.bio-techne.com/resources/instrument-software-download-center/compass-software-simple-western) in the Folder "raw data compass". Furthermore, we already added uncropped Lane views (folder "Revised Figures" for new Figures and "Lane view unaltered" for the complete Lane view) in the last submission as indicated in the files attached (compared to "Pre-Revision Figures").

File is attached in "Non-published Material".

We feel we adressed all comments accordingly.

Round 4

Reviewer 1 Report

"We did not perform Western Blotting, but Simple Western, which we clearly stated in the first and subsequent versions of the manuscript:"

Authors should perform WB again.

Author Response

As previously stated, we did not perform Western Blotting on these samples, so we can not perform it again, this would be another methodology we have to employ for an existing experiment. If direly needed, we could perform WB with our samples, although we do not think it is necessary since we evaluated both methodologies in our previous publication, https://onlinelibrary.wiley.com/doi/10.1111/acel.13914, Supp. Fig. 4, which we attached.
